# Flexible Models for Microclustering with Application to Entity Resolution

**Giacomo Zanella**[*]
Department of Decision Sciences
Bocconi University
giacomo.zanella@unibocconi.it

**Brenda Betancourt**[*]
Department of Statistical Science
Duke University
bb222@stat.duke.edu

**Hanna Wallach**
Microsoft Research
hanna@dirichlet.net

**Jeffrey Miller**
Department of Biostatistics
Harvard University
jwmiller@hsph.harvard.edu

**Abbas Zaidi**
Department of Statistical Science
Duke University
amz19@stat.duke.edu

**Rebecca C. Steorts**
Departments of Statistical Science and Computer Science
Duke University
beka@stat.duke.edu

## Abstract

Most generative models for clustering implicitly assume that the number of data points in each cluster grows linearly with the total number of data points. Finite mixture models, Dirichlet process mixture models, and Pitman–Yor process mixture models make this assumption, as do all other infinitely exchangeable clustering models. However, for some applications, this assumption is inappropriate. For example, when performing entity resolution, the size of each cluster should be unrelated to the size of the data set, and each cluster should contain a negligible fraction of the total number of data points. These applications require models that yield clusters whose sizes grow sublinearly with the size of the data set. We address this requirement by defining the microclustering property and introducing a new class of models that can exhibit this property. We compare models within this class to two commonly used clustering models using four entity-resolution data sets.

## 1 Introduction

Many clustering applications require models that assume cluster sizes grow linearly with the size of the data set. These applications include topic modeling, inferring population structure, and discriminating among cancer subtypes. Infinitely exchangeable clustering models, including finite mixture models, Dirichlet process mixture models, and Pitman–Yor process mixture models, all make this linear-growth assumption, and have seen numerous successes when used in these contexts. For other clustering applications, such as entity resolution, this assumption is inappropriate. Entity resolution (including record linkage and de-duplication) involves identifying duplicate[2] records in noisy databases [1, 2], traditionally by directly linking records to one another. Unfortunately, this traditional approach is computationally infeasible for large data sets—a serious limitation in "the age of big data" [1, 3]. As a

---

[*]Giacomo Zanella and Brenda Betancourt are joint first authors.

[2]In the entity resolution literature, the term "duplicate records" does not mean that the records are identical, but rather that the records are corrupted, degraded, or otherwise noisy representations of the same entity.

result, researchers increasingly treat entity resolution as a clustering problem, where each entity is implicitly associated with one or more records and the inference goal is to recover the latent entities (clusters) that correspond to the observed records (data points) [4, 5, 6]. In contrast to other clustering applications, the number of data points in each cluster should remain small, even for large data sets. Applications like this require models that yield clusters whose sizes grow sublinearly with the total number of data points [7]. To address this requirement, we define the microclustering property in section 2 and, in section 3, introduce a new class of models that can exhibit this property. In section 4, we compare two models within this class to two commonly used infinitely exchangeable clustering models.

## 2 The Microclustering Property

To cluster $N$ data points $x_1, \ldots, x_N$ using a partition-based Bayesian clustering model, one first places a prior over partitions of $[N] = \{1, \ldots, N\}$. Then, given a partition $C_N$ of $[N]$, one models the data points in each part $c \in C_N$ as jointly distributed according to some chosen distribution. Finally, one computes the posterior distribution over partitions and, e.g., uses it to identify probable partitions of $[N]$. Mixture models are a well-known type of partition-based Bayesian clustering model, in which $C_N$ is implicitly represented by a set of cluster assignments $z_1, \ldots, z_N$. These cluster assignments can be regarded as the first $N$ elements of an infinite sequence $z_1, z_2, \ldots$, drawn a priori from

$$\boldsymbol{\pi} \sim H \quad \text{and} \quad z_1, z_2, \ldots \mid \boldsymbol{\pi} \overset{\text{iid}}{\sim} \boldsymbol{\pi}, \tag{1}$$

where $H$ is a prior over $\boldsymbol{\pi}$ and $\boldsymbol{\pi}$ is a vector of mixture weights with $\sum_l \pi_l = 1$ and $\pi_l \geq 0$ for all $l$. Commonly used mixture models include (a) finite mixtures where the dimensionality of $\boldsymbol{\pi}$ is fixed and $H$ is usually a Dirichlet distribution; (b) finite mixtures where the dimensionality of $\boldsymbol{\pi}$ is a random variable [8, 9]; (c) Dirichlet process (DP) mixtures where the dimensionality of $\boldsymbol{\pi}$ is infinite [10]; and (d) Pitman–Yor process (PYP) mixtures, which generalize DP mixtures [11].

Equation 1 implicitly defines a prior over partitions of $\mathbb{N} = \{1, 2, \ldots\}$. Any random partition $C_{\mathbb{N}}$ of $\mathbb{N}$ induces a sequence of random partitions $(C_N : N = 1, 2, \ldots)$, where $C_N$ is a partition of $[N]$. Via the strong law of large numbers, the cluster sizes in any such sequence obtained via equation 1 grow linearly with $N$ because, with probability one, for all $l$, $\frac{1}{N} \sum_{n=1}^{N} I(z_n = l) \to \pi_l$ as $N \to \infty$, where $I(\cdot)$ denotes the indicator function. Unfortunately, this linear growth assumption is not appropriate for entity resolution and other applications that require clusters whose sizes grow sublinearly with $N$.

To address this requirement, we therefore define the microclustering property: A sequence of random partitions $(C_N : N = 1, 2, \ldots)$ exhibits the microclustering property if $M_N$ is $o_p(N)$, where $M_N$ is the size of the largest cluster in $C_N$, or, equivalently, if $M_N / N \to 0$ in probability as $N \to \infty$.

A clustering model exhibits the microclustering property if the sequence of random partitions implied by that model satisfies the above definition. No mixture model can exhibit the microclustering property (unless its parameters are allowed to vary with $N$). In fact, Kingman's paintbox theorem [12, 13] implies that any exchangeable partition of $\mathbb{N}$, such as a partition obtained using equation 1, is either equal to the trivial partition in which each part contains one element or satisfies $\liminf_{N \to \infty} M_N / N > 0$ with positive probability. By Kolmogorov's extension theorem, a sequence of random partitions $(C_N : N = 1, 2, \ldots)$ corresponds to an exchangeable random partition of $\mathbb{N}$ whenever (a) each $C_N$ is finitely exchangeable (i.e., its probability is invariant under permutations of $\{1, \ldots, N\}$) and (b) the sequence is projective (also known as consistent in distribution)—i.e., if $N' < N$, the distribution over $C_{N'}$ coincides with the marginal distribution over partitions of $[N']$ induced by the distribution over $C_N$. Therefore, to obtain a nontrivial model that exhibits the microclustering property, we must sacrifice either (a) or (b). Previous work [14] sacrificed (a); in this paper, we instead sacrifice (b).

Sacrificing finite exchangeability and sacrificing projectivity have very different consequences. If a partition-based Bayesian clustering model is not finitely exchangeable, then inference will depend on the order of the data points. For most applications, this consequence is undesirable—there is no reason to believe that the order of the data points is meaningful. In contrast, if a model lacks projectivity, then the implied joint distribution over a subset of the data points in a data set will not be the same as the joint distribution obtained by modeling the subset directly. In the context of entity resolution, sacrificing projectivity is a more natural and less restrictive choice than sacrificing finite exchangeability.

## 3    Kolchin Partition Models for Microclustering

We introduce a new class of Bayesian models for microclustering by placing a prior on the number of clusters $K$ and, given $K$, modeling the cluster sizes $N_1, \ldots, N_K$ directly. We start by defining

$$K \sim \boldsymbol{\kappa} \quad \text{and} \quad N_1, \ldots, N_K \mid K \overset{\text{iid}}{\sim} \boldsymbol{\mu}, \tag{2}$$

where $\boldsymbol{\kappa} = (\kappa_1, \kappa_2, \ldots)$ and $\boldsymbol{\mu} = (\mu_1, \mu_2, \ldots)$ are probability distributions over $\mathbb{N} = \{1, 2, \ldots\}$. We then define $N = \sum_{k=1}^{K} N_k$ and, given $N_1, \ldots, N_K$, generate a set of cluster assignments $z_1, \ldots, z_N$ by drawing a vector uniformly at random from the set of permutations of $(\underbrace{1, \ldots, 1}_{N_1 \text{ times}}, \underbrace{2, \ldots, 2}_{N_2 \text{ times}}, \ldots \ldots, \underbrace{K, \ldots, K}_{N_K \text{ times}})$. The cluster assignments $z_1, \ldots, z_N$ induce a random partition $C_N$ of $[N]$, where $N$ is itself a random variable—i.e., $C_N$ is a random partition of a random number of elements. We refer to the resulting class of marginal distributions over $C_N$ as Kolchin partition (KP) models [15, 16] because the form of equation 2 is closely related to Kolchin's representation theorem for Gibbs-type partitions (see, e.g., 16, theorem 1.2). For appropriate choices of $\boldsymbol{\kappa}$ and $\boldsymbol{\mu}$, KP models can exhibit the microclustering property (see appendix B for an example).

If $\mathscr{C}_N$ denotes the set of all possible partitions of $[N]$, then $\bigcup_{N=1}^{\infty} \mathscr{C}_N$ is the set of all possible partitions of $[N]$ for all $N \in \mathbb{N}$. The probability of any given partition $C_N \in \bigcup_{N=1}^{\infty} \mathscr{C}_N$ is

$$P(C_N) = \frac{|C_N|! \, \kappa_{|C_N|}}{N!} \left( \prod_{c \in C_N} |c|! \, \mu_{|c|} \right), \tag{3}$$

where $|\cdot|$ denotes the cardinality of a set, $|C_N|$ is the number of clusters in $C_N$, and $|c|$ is the number of elements in cluster c. In practice, however, $N$ is usually observed. Conditioned on $N$, a KP model implies that $P(C_N \mid N) \propto |C_N|! \, \kappa_{|C_N|} \left( \prod_{c \in C_N} |c|! \, \mu_{|c|} \right)$. Equation 3 leads to a "reseating algorithm"—much like the Chinese restaurant process (CRP)—derived by sampling from $P(C_N \mid N, C_N \setminus n)$, where $C_N \setminus n$ is the partition obtained by removing element $n$ from $C_N$:

- for $n = 1, \ldots, N$, reassign element $n$ to
  - an existing cluster $c \in C_N \setminus n$ with probability $\propto (|c| + 1) \frac{\mu_{(|c|+1)}}{\mu_{|c|}}$
  - or a new cluster with probability $\propto (|C_N \setminus n| + 1) \frac{\kappa_{(|C_N \setminus n|+1)}}{\kappa_{|C_N \setminus n|}} \mu_1$.

We can use this reseating algorithm to draw samples from $P(C_N \mid N)$; however, unlike the CRP, it does not produce an exact sample if it is used to incrementally construct a partition from the empty set. In practice, this limitation does not lead to any negative consequences because standard posterior inference sampling methods do not rely on this property. When a KP model is used as the prior in a partition-based clustering model—e.g., as an alternative to equation 1—the resulting Gibbs sampling algorithm for $C_N$ is similar to this reseating algorithm, but accompanied by likelihood terms. Unfortunately, this algorithm is slow for large data sets. In appendix C, we therefore propose a faster Gibbs sampling algorithm—the chaperones algorithm—that is particularly well suited to microclustering.

In sections 3.1 and 3.2, we introduce two related KP models for microclustering, and in section 3.4 we explain how KP models can be applied in the context of entity resolution with categorical data.

### 3.1    The NBNB Model

We start with equation 3 and define

$$\boldsymbol{\kappa} = \text{NegBin}\,(a, q) \quad \text{and} \quad \boldsymbol{\mu} = \text{NegBin}\,(r, p), \tag{4}$$

where $\text{NegBin}(a, q)$ and $\text{NegBin}(r, p)$ are negative binomial distributions truncated to $\mathbb{N} = \{1, 2, \ldots\}$. We assume that $a > 0$ and $q \in (0, 1)$ are fixed hyperparameters, while $r$ and $p$ are distributed as $r \sim \text{Gam}(\eta_r, s_r)$ and $p \sim \text{Beta}(u_p, v_p)$ for fixed $\eta_r$, $s_r$, $u_p$ and $v_p$.[3] We refer to the resulting marginal distribution over $C_N$ as the negative binomial–negative binomial (NBNB) model.

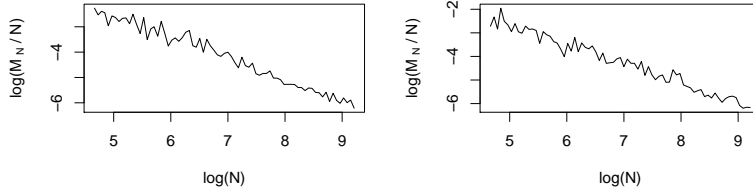

Figure 1: The NBNB (left) and NBD (right) models appear to exhibit the microclustering property.

By substituting equation 4 into equation 3, we obtain the probability of $C_N$ conditioned $N$:

$$P(C_N \mid N, a, q, r, p) \propto \Gamma\left(|C_N| + a\right) \beta^{|C_N|} \prod_{c \in C_N} \frac{\Gamma\left(|c| + r\right)}{\Gamma\left(r\right)}, \tag{5}$$

where $\beta = \frac{q\,(1-p)^r}{1-(1-p)^r}$. We provide the complete derivation of equation 5, along with the conditional posterior distributions over $r$ and $p$, in appendix A.2. Posterior inference for the NBNB model involves alternating between (a) sampling $C_N$ from $P(C_N \mid N, a, q, r, p)$ using the chaperones algorithm and (b) sampling $r$ and $p$ from their respective conditional posteriors using, e.g., slice sampling [17].

## 3.2   The NBD Model

Although $\boldsymbol{\kappa} = \text{NegBin}\,(a, q)$ will yield plausible values of $K$, $\boldsymbol{\mu} = \text{NegBin}\,(r, p)$ may not be sufficiently flexible to capture realistic properties of $N_1, \ldots, N_K$, especially when $K$ is large. For example, in a record-linkage application involving two otherwise noise-free databases containing thousands of records, $K$ will be large and each $N_k$ will be at most two. A negative binomial distribution cannot capture this property. We therefore define a second KP model—the negative binomial–Dirichlet (NBD) model—by taking a nonparametric approach to modeling $N_1, \ldots, N_K$ and drawing $\boldsymbol{\mu}$ from an infinite-dimensional Dirichlet distribution over the positive integers:

$$\boldsymbol{\kappa} = \text{NegBin}\,(a, q) \quad \text{and} \quad \boldsymbol{\mu} \mid \alpha, \boldsymbol{\mu}^{(0)} \sim \text{Dir}\left(\alpha, \boldsymbol{\mu}^{(0)}\right), \tag{6}$$

where $\alpha > 0$ is a fixed concentration parameter and $\boldsymbol{\mu}^{(0)} = (\mu_1^{(0)}, \mu_2^{(0)}, \cdots)$ is a fixed base measure with $\sum_{m=1}^{\infty} \mu_m^{(0)} = 1$ and $\mu_m^{(0)} \geq 0$ for all $m$. The probability of $C_N$ conditioned on $N$ and $\boldsymbol{\mu}$ is

$$P(C_N \mid N, a, q, \boldsymbol{\mu}) \propto \Gamma\left(|C_N| + a\right) q^{|C_N|} \prod_{c \in C_N} |c|!\, \mu_{|c|}. \tag{7}$$

Posterior inference for the NBD model involves alternating between (a) sampling $C_N$ from $P(C_N \mid N, a, q, \boldsymbol{\mu})$ using the chaperones algorithm and (b) sampling $\boldsymbol{\mu}$ from its conditional posterior:

$$\boldsymbol{\mu} \mid C_N, \alpha, \boldsymbol{\mu}^{(0)} \sim \text{Dir}\left(\alpha\,\mu_1^{(0)} + L_1, \alpha\,\mu_2^{(0)} + L_2, \ldots\right), \tag{8}$$

where $L_m$ is the number of clusters of size $m$ in $C_N$. Although $\boldsymbol{\mu}$ is an infinite-dimensional vector, only the first $N$ elements affect $P(C_N \mid a, q, \boldsymbol{\mu})$. Therefore, it is sufficient to sample the $(N+1)$-dimensional vector $(\mu_1, \ldots, \mu_N, 1 - \sum_{m=1}^{N} \mu_m)$ from equation 8, modified accordingly, and retain only $\mu_1, \ldots, \mu_N$. We provide complete derivations of equations 7 and 8 in appendix A.3.

## 3.3   The Microclustering Property for the NBNB and NBD Models

Figure 1 contains empirical evidence suggesting that the NBNB and NBD models both exhibit the microclustering property. For each model, we generated samples of $M_N / N$ for $N = 100, \ldots, 10^4$. For the NBNB model, we set $a = 1$, $q = 0.5$, $r = 1$, and $p = 0.5$ and generated the samples using rejection sampling. For the NBD model, we set $a = 1$, $q = 0.5$, and $\alpha = 1$ and set $\boldsymbol{\mu}^{(0)}$ to be a geometric distribution over $\mathbb{N} = \{1, 2, \ldots\}$ with a parameter of 0.5. We generated the samples using MCMC methods. For both models, $M_N / N$ appears to converge to zero in probability as $N \to \infty$, as desired.

In appendix B, we also prove that a variant of the NBNB model exhibits the microclustering property.

### 3.4 Application to Entity Resolution

KP models can be used to perform entity resolution. In this context, the data points $x_1, \ldots, x_N$ are observed records and the $K$ clusters are latent entities. If each record consists of $F$ categorical fields, then

$$C_N \sim \text{KP model} \tag{9}$$

$$\boldsymbol{\theta}_{fk} \mid \delta_f, \boldsymbol{\gamma}_f \sim \text{Dir}\left(\delta_f, \boldsymbol{\gamma}_f\right) \tag{10}$$

$$z_n \sim \zeta(C_N, n) \tag{11}$$

$$x_{fn} \mid z_n, \boldsymbol{\theta}_{f1}, \ldots, \boldsymbol{\theta}_{fK} \sim \text{Cat}\left(\boldsymbol{\theta}_{fz_n}\right) \tag{12}$$

for $f = 1, \ldots, F$, $k = 1, \ldots, K$, and $n = 1, \ldots, N$, where $\zeta(C_N, n)$ maps the $n^{\text{th}}$ record to a latent cluster assignment $z_n$ according to $C_N$. We assume that $\delta_f > 0$ is distributed as $\delta_f \sim \text{Gam}(1, 1)$, while $\boldsymbol{\gamma}_f$ is fixed. Via Dirichlet–multinomial conjugacy, we can marginalize over $\boldsymbol{\theta}_{11}, \ldots, \boldsymbol{\theta}_{FK}$ to obtain a closed-form expression for $P(x_1, \ldots, x_N \mid z_1, \ldots, z_N, \delta_f, \boldsymbol{\gamma}_f)$. Posterior inference involves alternating between (a) sampling $C_N$ from $P(C_N \mid x_1, \ldots, x_N, \delta_f)$ using the chaperones algorithm accompanied by appropriate likelihood terms, (b) sampling the parameters of the KP model from their conditional posteriors, and (c) sampling $\delta_f$ from its conditional posterior using slice sampling.

## 4 Experiments

In this section, we compare two entity resolution models based on the NBNB model and the NBD model to two similar models based on the DP mixture model [10] and the PYP mixture model [11]. All four models use the likelihood in equations 10 and 12. For the NBNB model and the NBD model, we set $a$ and $q$ to reflect a weakly informative prior belief that $\mathbb{E}[K] = \sqrt{\text{Var}[K]} = \frac{N}{2}$. For the NBNB model, we set $\eta_r = s_r = 1$ and $u_p = v_p = 2$.[4] For the NBD model, we set $\alpha = 1$ and set $\boldsymbol{\mu}^{(0)}$ to be a geometric distribution over $\mathbb{N} = \{1, 2, \ldots\}$ with a parameter of 0.5. This base measure reflects a prior belief that $\mathbb{E}[N_k] = 2$. Finally, to ensure a fair comparison between the two different classes of model, we set the DP and PYP concentration parameters to reflect a prior belief that $\mathbb{E}[K] = \frac{N}{2}$.

We assess how well each model "fits" four data sets typical of those arising in real-world entity resolution applications. For each data set, we consider four statistics: (a) the number of singleton clusters, (b) the maximum cluster size, (c) the mean cluster size, and (d) the $90^{\text{th}}$ percentile of cluster sizes. We compare each statistic's true value to its posterior distribution according to each of the models. For each model and data set combination, we also consider five entity-resolution summary statistics: (a) the posterior expected number of clusters, (b) the posterior standard error, (c) the false negative rate, (d) the false discovery rate, and (e) the posterior expected value of $\delta_f = \delta$ for $f = 1, \ldots, F$. The false negative and false discovery rates are both invariant under permutations of $1, \ldots, K$ [5, 18].

### 4.1 Data Sets

We constructed four realistic data sets, each consisting of $N$ records associated with $K$ entities.

**Italy:** We derived this data set from the Survey on Household Income and Wealth, conducted by the Bank of Italy every two years. There are nine categorical fields, including year of birth, employment status, and highest level of education attained. Ground truth is available via unique identifiers based upon social security numbers; roughly 74% of the clusters are singletons. We used the 2008 and 2010 databases from the Fruili region to create a record-linkage data set consisting of $N = 789$ records; each $N_k$ is at most two. We discarded the records themselves, but preserved the number of fields, the empirical distribution of categories for each field, the number of clusters, and the cluster sizes. We then generated synthetic records using equations 10 and 12. We created three variants of this data set, corresponding to $\delta = 0.02, 0.05, 0.1$. For all three, we used the empirical distribution of categories for field $f$ as $\boldsymbol{\gamma}_f$. By generating synthetic records in this fashion, we preserve the pertinent characteristics of the original data, while making it easy to isolate the impacts of the different priors over partitions.

**NLTCS5000:** We derived this data set from the National Long Term Care Survey (NLTCS)[5]—a longitudinal survey of older Americans, conducted roughly every six years. We used four of the

available fields: date of birth, sex, state of residence, and regional office. We split date of birth into three separate fields: day, month, and year. Ground truth is available via social security numbers; roughly 68% of the clusters are singletons. We used the 1982, 1989, and 1994 databases and down-sampled the records, preserving the proportion of clusters of each size and the maximum cluster size, to create a record-linkage data set of $N = 5,000$ records; each $N_k$ is at most three. We then generated synthetic records using the same approach that we used to create the Italy data set.

**Syria2000 and SyriaSizes:** We constructed these data sets from data collected by four human-rights groups between 2011 and 2014 on people killed in the Syrian conflict [19, 20]. Hand-matched ground truth is available from the Human Rights Data Analysis Group. Because the records were hand matched, the data are noisy and potentially biased. Performing entity resolution is non-trivial because there are only three categorical fields: gender, governorate, and date of death. We split date of death, which is present for most records, into three separate fields: day, month, and year. However, because the records only span four years, the year field conveys little information. In addition, most records are male, and there are only fourteen governorates. We created the Syria2000 data set by down-sampling the records, preserving the proportion of clusters of each size, to create a data set of $N = 2,000$ records; the maximum cluster size is five. We created the SyriaSizes data set by down-sampling the records, preserving some of the larger clusters (which necessarily contain within-database duplications), to create a data set of $N = 6,700$ records; the maximum cluster size is ten. We provide the empirical distribution over cluster sizes for each data set in appendix D. We generated synthetic records for both data sets using the same approach that we used to create the Italy data set.

## 4.2  Results

We report the results of our experiments in table 1 and figure 2. The NBNB and NBD models outperformed the DP and PYP models for almost all variants of the Italy and NLTCS5000 data sets. In general, the NBD model performed the best of the four, and the differences between the models' performance grew as the value of $\delta$ increased. For the Syria2000 and SyriaSizes data sets, we see no consistent pattern to the models' abilities to recover the true values of the data-set statistics. Moreover, all four models had poor false negative rates, and false discovery rates—most likely because these data sets are extremely noisy and contain very few fields. We suspect that no entity resolution model would perform well for these data sets. For three of the four data sets, the exception being the Syria2000 data set, the DP model and the PYP model both greatly overestimated the number of clusters for larger values of $\delta$. Taken together, these results suggest that the flexibility of the NBNB and NBD models make them more appropriate choices for most entity resolution applications.

## 5  Summary

Infinitely exchangeable clustering models assume that cluster sizes grow linearly with the size of the data set. Although this assumption is reasonable for some applications, it is inappropriate for others. For example, when entity resolution is treated as a clustering problem, the number of data points in each cluster should remain small, even for large data sets. Applications like this require models that yield clusters whose sizes grow sublinearly with the size of the data set. We introduced the microclustering property as one way to characterize models that address this requirement. We then introduced a highly flexible class of models—KP models—that can exhibit this property. We presented two models within this class—the NBNB model and the NBD model—and showed that they are better suited to entity resolution applications than two infinitely exchangeable clustering models. We therefore recommend KP models for applications where the size of each cluster should be unrelated to the size of the data set, and each cluster should contain a negligible fraction of the total number of data points.

**Acknowledgments**

We thank Tamara Broderick, David Dunson, Merlise Clyde, and Abel Rodriguez for conversations that helped form the ideas in this paper. In particular, Tamara Broderick played a key role in developing the idea of microclustering. We also thank the Human Rights Data Analysis Group for providing us with data. This work was supported in part by NSF grants SBE-0965436, DMS-1045153, and IIS-1320219; NIH grant 5R01ES017436-05; the John Templeton Foundation; the Foerster-Bernstein Postdoctoral Fellowship; the UMass Amherst CIIR; and an EPSRC Doctoral Prize Fellowship.

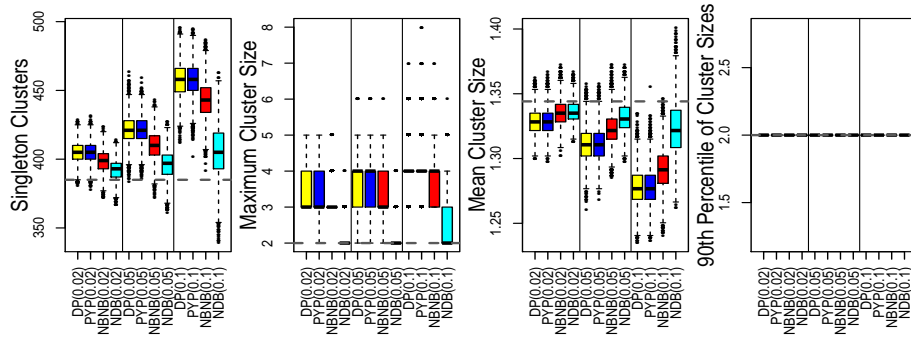

(a) Italy: NBD model > NBNB model > PYP mixture model > DP mixture model.

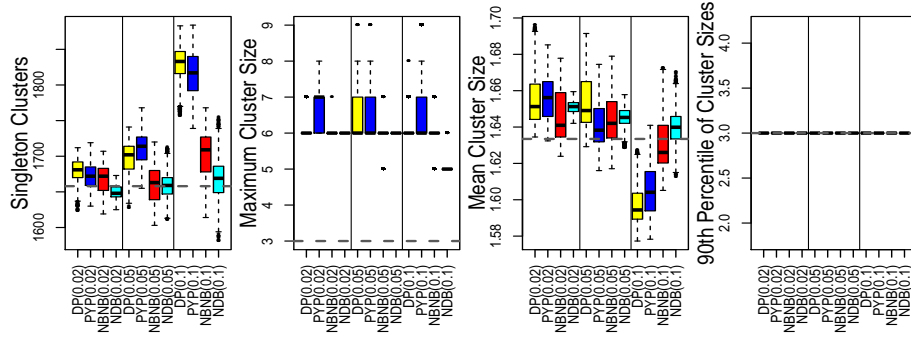

(b) NLTCS5000: NBD model > NBNB model > PYP mixture model > DP mixture model.

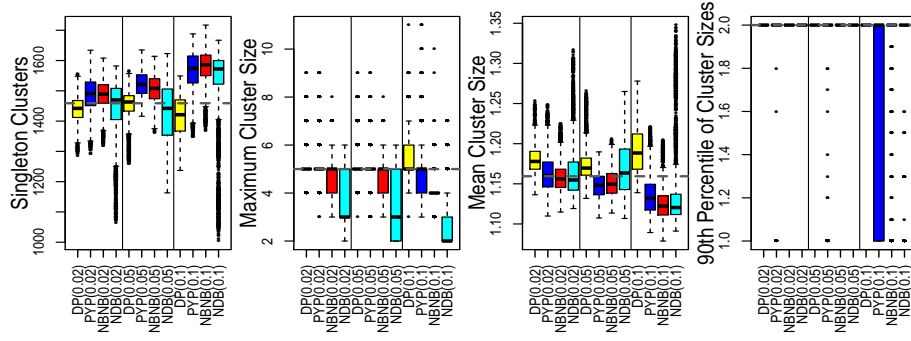

(c) Syria2000: the models perform similarly because there are so few fields.

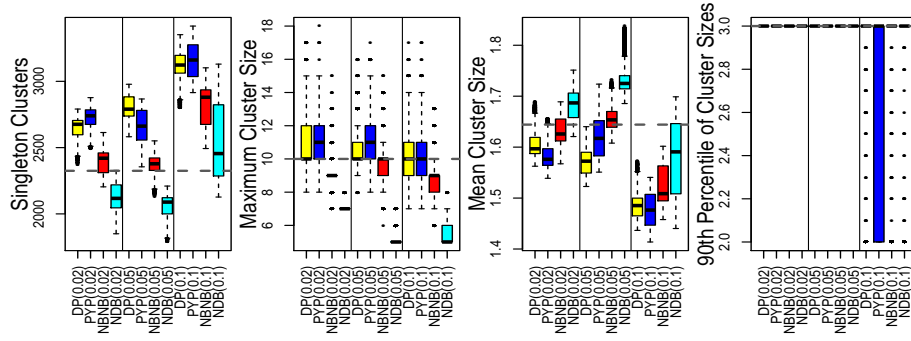

(d) SyriaSizes: the models perform similarly because there are so few fields.

Figure 2: Box plots depicting the true value (dashed line) of each data-set statistic for each variant of each data set, as well as its posterior distribution according to each of the four entity resolution models.

Table 1: Entity-resolution summary statistics—the posterior expected number of clusters, the posterior standard error, the false negative rate (lower is better), the false discovery rate (lower is better), and the posterior expected value of $\delta$—for each variant of each data set and each of the four models.

| Data Set | True $K$ | Variant | Model | $\mathbb{E}[K]$ | Std. Err. | FNR | FDR | $\mathbb{E}[\delta]$ |
|---|---|---|---|---|---|---|---|---|
| Italy | 587 | $\delta = 0.02$ | DP | 594.00 | 4.51 | 0.07 | 0.03 | 0.02 |
| | | | PYP | 593.90 | 4.52 | 0.07 | 0.03 | 0.02 |
| | | | NBNB | 591.00 | 4.43 | 0.04 | 0.03 | 0.02 |
| | | | NBD | 590.50 | 3.64 | 0.03 | 0.00 | 0.02 |
| | | $\delta = 0.05$ | DP | 601.60 | 5.89 | 0.13 | 0.03 | 0.03 |
| | | | PYP | 601.50 | 5.90 | 0.13 | 0.03 | 0.04 |
| | | | NBNB | 596.40 | 5.79 | 0.11 | 0.04 | 0.04 |
| | | | NBD | 592.60 | 5.20 | 0.09 | 0.04 | 0.04 |
| | | $\delta = 0.1$ | DP | 617.40 | 7.23 | 0.27 | 0.06 | 0.07 |
| | | | PYP | 617.40 | 7.22 | 0.27 | 0.05 | 0.07 |
| | | | NBNB | 610.90 | 7.81 | 0.24 | 0.06 | 0.08 |
| | | | NBD | 596.60 | 9.37 | 0.18 | 0.05 | 0.10 |
| NLTCS5000 | 3,061 | $\delta = 0.02$ | DP | 3021.70 | 24.96 | 0.02 | 0.11 | 0.03 |
| | | | PYP | 3018.70 | 25.69 | 0.03 | 0.11 | 0.03 |
| | | | NBNB | 3037.80 | 25.18 | 0.02 | 0.07 | 0.02 |
| | | | NBD | 3028.20 | 5.65 | 0.01 | 0.09 | 0.03 |
| | | $\delta = 0.05$ | DP | 3024.00 | 26.15 | 0.05 | 0.13 | 0.06 |
| | | | PYP | 3045.80 | 23.66 | 0.05 | 0.10 | 0.05 |
| | | | NBNB | 3040.90 | 24.86 | 0.04 | 0.06 | 0.05 |
| | | | NBD | 3039.30 | 10.17 | 0.03 | 0.07 | 0.06 |
| | | $\delta = 0.1$ | DP | 3130.50 | 21.44 | 0.12 | 0.09 | 0.10 |
| | | | PYP | 3115.10 | 25.73 | 0.13 | 0.10 | 0.10 |
| | | | NBNB | 3067.30 | 25.31 | 0.11 | 0.08 | 0.11 |
| | | | NBD | 3049.10 | 16.48 | 0.09 | 0.08 | 0.12 |
| Syria2000 | 1,725 | $\delta = 0.02$ | DP | 1695.20 | 25.40 | 0.70 | 0.27 | 0.07 |
| | | | PYP | 1719.70 | 36.10 | 0.71 | 0.26 | 0.04 |
| | | | NBNB | 1726.80 | 27.96 | 0.70 | 0.28 | 0.05 |
| | | | NBD | 1715.20 | 51.56 | 0.67 | 0.28 | 0.02 |
| | | $\delta = 0.05$ | DP | 1701.80 | 31.15 | 0.77 | 0.31 | 0.07 |
| | | | PYP | 1742.90 | 24.33 | 0.75 | 0.32 | 0.04 |
| | | | NBNB | 1738.30 | 25.48 | 0.74 | 0.31 | 0.04 |
| | | | NBD | 1711.40 | 47.10 | 0.69 | 0.32 | 0.03 |
| | | $\delta = 0.1$ | DP | 1678.10 | 40.56 | 0.81 | 0.19 | 0.18 |
| | | | PYP | 1761.20 | 39.38 | 0.81 | 0.22 | 0.08 |
| | | | NBNB | 1779.40 | 29.84 | 0.77 | 0.26 | 0.04 |
| | | | NBD | 1757.30 | 73.60 | 0.74 | 0.25 | 0.03 |
| SyriaSizes | 4,075 | $\delta = 0.02$ | DP | 4175.70 | 66.04 | 0.65 | 0.17 | 0.01 |
| | | | PYP | 4234.30 | 68.55 | 0.64 | 0.19 | 0.01 |
| | | | NBNB | 4108.70 | 70.56 | 0.65 | 0.19 | 0.01 |
| | | | NBD | 3979.50 | 70.85 | 0.68 | 0.20 | 0.03 |
| | | $\delta = 0.05$ | DP | 4260.00 | 77.18 | 0.71 | 0.21 | 0.02 |
| | | | PYP | 4139.10 | 104.22 | 0.75 | 0.18 | 0.04 |
| | | | NBNB | 4047.10 | 55.18 | 0.73 | 0.20 | 0.04 |
| | | | NBD | 3863.90 | 68.05 | 0.75 | 0.22 | 0.07 |
| | | $\delta = 0.1$ | DP | 4507.40 | 82.27 | 0.80 | 0.19 | 0.03 |
| | | | PYP | 4540.30 | 100.53 | 0.80 | 0.20 | 0.03 |
| | | | NBNB | 4400.60 | 111.91 | 0.80 | 0.23 | 0.03 |
| | | | NBD | 4251.90 | 203.23 | 0.82 | 0.25 | 0.04 |

## Footnotes

[3] We use the shape-and-rate parameterization of the gamma distribution.

[4]We used $p \sim \text{Beta}(2, 2)$ because a uniform prior implies an unrealistic prior belief that $\mathbb{E}[N_k] = \infty$.

[5]`http://www.nltcs.aas.duke.edu/`

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
