[Supplementary Material]

# Supplementary Material for "Flexible Models for Microclustering with Application to Entity Resolution"

**Giacomo Zanella**[*]
Department of Decision Sciences
Bocconi University
giacomo.zanella@unibocconi.it

**Brenda Betancourt**[*]
Department of Statistical Science
Duke University
bb222@stat.duke.edu

**Hanna Wallach**
Microsoft Research
hanna@dirichlet.net

**Jeffrey Miller**
Department of Biostatistics
Harvard University
jwmiller@hsph.harvard.edu

**Abbas Zaidi**
Department of Statistical Science
Duke University
amz19@stat.duke.edu

**Rebecca C. Steorts**
Departments of Statistical Science and Computer Science
Duke University
beka@stat.duke.edu

## A   Derivation of $P(C_N)$

In this appendix, we derive $P(C_N)$ for a general KP model, as well as the NBNB and NBD models.

### A.1   KP Models

We start with equation 2 and note that

$$P(C_N) = P(C_N \mid K) \, P(K), \tag{1}$$

where $K = |C_N|$. To evaluate $P(C_N \mid K)$, we need to sum over all possible cluster assignments:

$$P(C_N \mid K) = \sum_{z_1,\ldots,z_N \in [K]} \underbrace{P(C_N \mid z_1,\ldots,z_N,K)}_{I(z_1,\ldots,z_N \Rightarrow C_N)} \, P(z_1,\ldots,z_N \mid K). \tag{2}$$

Since $N_1,\ldots,N_K$ are completely determined by $K$ and $z_1,\ldots,z_N$, it follows that

$$P(z_1,\ldots,z_N \mid K) = P(z_1,\ldots,z_N \mid N_1,\ldots,N_K,K) \, P(N_1,\ldots,N_K \mid K) \tag{3}$$

$$= \frac{\prod_{k=1}^{K} N_k!}{N!} \prod_{k=1}^{K} P(N_k \mid K) \tag{4}$$

$$= \frac{1}{N!} \prod_{k=1}^{K} N_k! \, \mu_{N_k}. \tag{5}$$

---

[*]Giacomo Zanella and Brenda Betancourt are joint first authors.

Therefore,

$$P(C_N \mid K) = \sum_{z_1,\ldots,z_N \in [K]} I(z_1,\ldots,z_N \Rightarrow C_N) \frac{1}{N!} \prod_{k=1}^{K} N_k! \, \mu_{N_k} \tag{6}$$

$$= \frac{1}{N!} \left( \prod_{c \in C_N} |c|! \, \mu_{|c|} \right) \sum_{z_1,\ldots,z_N \in [K]} I(z_1,\ldots,z_N \Rightarrow C_N) \tag{7}$$

$$= \frac{K!}{N!} \prod_{c \in C_N} |c|! \, \mu_{|c|}. \tag{8}$$

Substituting equation 8 into equation 1 and using $K \sim \kappa$ we obtain

$$P(C_N) = \frac{|C_N|! \, \kappa_{|C_N|}}{N!} \left( \prod_{c \in C_N} |c|! \, \mu_{|c|} \right). \tag{9}$$

## A.2 The NBNB Model

For fixed values of $r$ and $p$, the NBNB model is a specific case of a KP model with

$$\kappa_k = \frac{\Gamma(k+a) \, q^k \, (1-q)^a}{(1-(1-q)^a) \, \Gamma(a) \, k!} \quad \text{and} \quad \mu_m = \frac{\Gamma(m+r) \, p^m \, (1-p)^r}{(1-(1-p)^r) \, \Gamma(r) \, m!}, \tag{10}$$

for $k$ and $m$ in $\mathcal{N} = \{1, 2, \ldots\}$. Combining equations 9 and 10 gives

$$P(C_N \mid a, q, r, p) = \frac{|C_N|!}{N!} \frac{\Gamma(|C_N|+a) \, q^{|C_N|} \, (1-q)^a}{(1-(1-q)^a) \, \Gamma(a) \, |C_N|!} \prod_{c \in C_N} |c|! \frac{\Gamma(|c|+r) \, p^{|c|} \, (1-p)^r}{(1-(1-p)^r) \, \Gamma(r) \, |c|!} \tag{11}$$

$$= \frac{\Gamma(|C_N|+a) \, q^{|C_N|} \, (1-q)^a}{N! \, (1-(1-q)^a) \, \Gamma(a)} \prod_{c \in C_N} \frac{\Gamma(|c|+r) \, p^{|c|} \, (1-p)^r}{(1-(1-p)^r) \, \Gamma(r)}. \tag{12}$$

Conditioning on $N$ and removing constant terms, we obtain

$$P(C_N \mid N, a, q, r, p) \propto \Gamma(|C_N|+a) \, \beta^{|C_N|} \prod_{c \in C_N} \frac{\Gamma(|c|+r)}{\Gamma(r)}, \tag{13}$$

where $\beta = \frac{q\,(1-p)^r}{1-(1-p)^r}$. Equation 13 leads to the following reseating algorithm:

- for $n = 1, \ldots, N$, reassign element $n$ to
  - an existing cluster $c \in C_N \setminus n$ with probability $\propto |c| + r$
  - or a new cluster with probability $\propto (|C_N \setminus n| + a) \, \beta$.

Adding the prior terms for $r$ and $p$ to equation 12 we obtain the joint distribution of $C_N$, $r$ and $p$:

$$\begin{aligned}
&P(C_N, r, p \mid a, q, \eta_r, s_r, u_p, v_p) \\
&= P(r \mid \eta_r, s_r) \, P(p \mid u_p, v_p) \, P(C_N \mid r, p)
\end{aligned} \tag{14}$$

$$\begin{aligned}
&= \frac{r^{\eta_r - 1} e^{-\frac{r}{s_r}}}{\Gamma(\eta_r) \, s_r^{\eta_r}} \frac{p^{u_p - 1} (1-p)^{v_p - 1}}{B(u_p, v_p)} \times \\
&\quad \frac{\Gamma(|C_N|+a) \, q^{|C_N|} \, (1-q)^a}{N! \, (1-(1-q)^a) \, \Gamma(a)} \prod_{c \in C_N} \frac{\Gamma(|c|+r) \, p^{|c|} \, (1-p)^r}{(1-(1-p)^r) \, \Gamma(r)}
\end{aligned} \tag{15}$$

$$\begin{aligned}
&\propto r^{\eta_r - 1} \, e^{-\frac{r}{s_r}} \, p^{N + u_p - 1} \, (1-p)^{v_p - 1} \left( \frac{q\,(1-p)^r}{1-(1-p)^r} \right)^{|C_N|} \times \\
&\quad \frac{\Gamma(|C_N|+a)}{N!} \prod_{c \in C_N} \frac{\Gamma(|c|+r)}{\Gamma(r)}.
\end{aligned} \tag{16}$$

Therefore, the conditional posterior distributions over $r$ and $p$ are

$$P(r \mid C_N, p, \eta_r, s_r) \propto \frac{r^{\eta_r - 1} e^{-\frac{r}{s_r}} (1-p)^{r |C_N|}}{(1 - (1-p)^r)^{|C_N|}} \prod_{c \in C_N} \frac{\Gamma(|c| - 1 + r)}{\Gamma(r)} \tag{17}$$

$$P(p \mid C_N, r, u_p, v_p) \propto \frac{p^{N + u_p - 1}(1-p)^{r |C_N| + v_p - 1}}{(1 - (1-p)^r)^{|C_N|}}. \tag{18}$$

### A.3 The NBD Model

For fixed $\boldsymbol{\mu}$, the NBD model is a specific case of a KP model. Therefore,

$$P(C_N \mid a, q, \boldsymbol{\mu}) = \frac{\Gamma(|C_N| + a) \, q^{|C_N|} \, (1-q)^a}{N! \, (1 - (1-q)^a) \Gamma(a)} \prod_{c \in C_N} |c|! \, \mu_{|c|}. \tag{19}$$

Conditioning on $N$ and removing constant terms, we obtain

$$P(C_N \mid N, a, q, \boldsymbol{\mu}) \propto \Gamma(|C_N| + a) \, q^{|C_N|} \prod_{c \in |C_N|} |c|! \, \mu_{|c|}.$$

Via Dirichlet–multinomial conjugacy,

$$\boldsymbol{\mu} \mid C_N, \alpha, \boldsymbol{\mu}^{(0)} \sim \mathrm{Dir}\left(\alpha \, \mu_1^{(0)} + L_1, \alpha \, \mu_2^{(0)} + L_2, \dots\right), \tag{20}$$

where $L_m$ is the number of clusters of size $m$ in $C_N$. Although $\boldsymbol{\mu}$ is an infinite-dimensional vector, only the first $N$ elements affect $P(C_N \mid a, q, \boldsymbol{\mu})$. Therefore, it is sufficient to sample the $(N + 1)$-dimensional vector $(\mu_1, \dots, \mu_N, 1 - \sum_{m=1}^{N} \mu_m)$ from equation 20, modified accordingly:

$$(\mu_1, \dots, \mu_N, 1 - \sum_{m=1}^{N} \mu_m) \mid C_N, \alpha, \mu_1^{(0)}, \dots, \mu_N^{(0)}$$

$$\sim \mathrm{Dir}\left(\alpha \, \mu_1^{(0)} + L_1, \dots, \alpha \, \mu_N^{(0)} + L_N, \alpha\left(1 - \sum_{m=1}^{N} \mu_m^{(0)}\right)\right). \tag{21}$$

We can then discard $1 - \sum_{m=1}^{N} \mu_m$.

## B  Proof of the Microclustering Property for a Variant of the NBNB Model

**Theorem 1.** *If $C_N$ is drawn from a KP model with $\boldsymbol{\kappa} = NegBin\,(a, q)$ and $\boldsymbol{\mu} = NegBin\,(r, p)$,[2] then for all $\epsilon > 0$, $P(M_N / N \geq \epsilon) \to 0$ as $N \to \infty$, where $M_N$ is the size of the largest cluster in $C_N$.*

In this appendix, we provide a proof of theorem 1.

We use the following fact: $\Gamma(x + a) / \Gamma(x) \asymp x^a$ as $x \to \infty$ for any $a \in \mathbb{R}$ via Stirling's approximation to the gamma function. We use $f(x) \asymp g(x)$ to denote that $f(x) / g(x) \to 1$ as $x \to \infty$.

**Lemma 1.** *For any $k \in \{1, 2, \dots\}$, $P(K = k \mid N = n) \to 0$ as $n \to \infty$.*

*Proof.* Because $N \mid K = k \sim NegBin\,(kr, p)$,

$$P(K = k, N = n) = \frac{\Gamma(k + a)}{k! \, \Gamma(a)} (1 - q)^a \, q^k \, \frac{\Gamma(n + kr)}{n! \, \Gamma(kr)} (1 - p)^{kr} \, p^n.$$

Via the fact noted above, $\Gamma(n + kr) / \Gamma(n + kr + r) \asymp 1 / (n + kr)^r \to 0$ as $n \to \infty$, so

$$\frac{P(K = k, N = n)}{P(K = k + 1, N = n)} = \frac{\Gamma(k + a)(k + 1)}{\Gamma(k + a + 1) q} \frac{\Gamma(kr + r)}{\Gamma(kr)} \frac{\Gamma(n + kr)}{\Gamma(n + kr + r)} \to 0 \text{ as } n \to \infty.$$

Therefore,

$$P(K = k \mid N = n) = \frac{P(K = k, N = n)}{\sum_{k'=0}^{\infty} P(K = k', N = n)} \leq \frac{P(K = k, N = n)}{P(K = k + 1, N = n)} \to 0.$$

$\square$

**Lemma 2.** *For any $\epsilon \in (0,1)$, there exist $c_1, c_2, \ldots \geq 0$, not depending on $n$, such that $c_k \to 0$ as $k \to \infty$ and $k\, P(N_1 \,/\, n \geq \epsilon \,|\, K = k, N = n) \leq c_k$ for all $n \geq 2\,/\,\epsilon$ and $k \in \{1, 2, \ldots\}$.*

Before proving lemma 2, we first show how theorem 1 follows from it.

*Proof of theorem 1.* Let $\epsilon \in (0,1)$ and choose $c_1, c_2, \ldots$ by lemma 2. For any $n \geq 2\,/\,\epsilon$,

$$P(M_n\,/\,n \geq \epsilon \,|\, N = n)$$

$$= \sum_{k=1}^{\infty} P(N_1\,/\,n \geq \epsilon \text{ or } \cdots \text{ or } N_K\,/\,n \geq \epsilon \,|\, K = k, N = n)\, P(K = k \,|\, N = n)$$

$$\leq \sum_{k=1}^{\infty} \sum_{i=1}^{k} P(N_i\,/\,n \geq \epsilon \,|\, K = k, N = n)\, P(K = k \,|\, N = n)$$

$$= \sum_{k=1}^{\infty} k\, P(N_1\,/\,n \geq \epsilon \,|\, K = k, N = n)\, P(K = k \,|\, N = n)$$

$$\leq \sum_{k=1}^{\infty} c_k\, P(K = k \,|\, N = n) \leq \sup\{c_k : k > m\} + \sum_{k=1}^{m} c_k\, P(K = k \,|\, N = n)$$

for any $m \geq 1$. (We note that we only summed over $k \geq 1$ because $P(K = 0 \,|\, N = n) = 0$ for any $n \geq 1$.) Therefore, via lemma 1, $\limsup_n P(M_n\,/\,n \geq \epsilon \,|\, N = n) \leq \sup\{c_k : k > m\}$. Finally, because $\sup\{c_k : k > m\} \to 0$ as $m \to \infty$, theorem 1 follows directly from lemma 2, as desired. $\quad\square$

To prove lemma 2, we need two supporting results.

**Lemma 3.** *If $b > (r+1)\,/\,r$ and $\theta_k \sim Beta\,(r, (k-1)\,r)$, then $k\, P(\theta_k \geq \frac{b \log(k)}{k}) \to 0$ as $k \to \infty$.*

*Proof.* Let $a_k = (b \log(k))\,/\,k$, and suppose that $k$ is large enough that $a_k \in (0,1)$. First, for any $\theta \in (a_k, 1)$, we have $\theta^{r-1} \leq 1\,/\,a_k$. Second, $B\,(r, (k-1)\,r) = \Gamma(r)\,\Gamma(kr - r)\,/\,\Gamma(kr) \asymp \Gamma(r)\,(kr)^{-r}$ as $k \to \infty$, via Stirling's approximation, as we noted previously. Third, because $1 + x \leq \exp(x)$ for any $x \in \mathbb{R}$, $(1 - a_k)^{kr} \leq \exp(-a_k)^{kr} = k^{-rb}$. Therefore, we obtain

$$k\, P(\theta_k \geq a_k)$$

$$= \frac{k}{B\,(r, (k-1)\,r)} \int_{a_k}^{1} \theta^{r-1}\,(1-\theta)^{(k-1)\,r-1}\, \mathrm{d}\theta$$

$$\leq \frac{k\,/\,a_k}{B\,(r, (k-1)\,r)} \int_{a_k}^{1} (1-\theta)^{(k-1)r-1}\, \mathrm{d}\theta = \frac{k\,/\,a_k}{B\,(r, (k-1)\,r)} \frac{(1 - a_k)^{(k-1)\,r}}{(k-1)\,r}$$

$$\leq \frac{k\,/\,a_k}{B\,(r, (k-1)\,r)} \frac{k^{-rb}\,(1-a_k)^{-r}}{(k-1)\,r} \asymp \frac{k^2\,/\,(b\log(k))}{\Gamma(r)\,(kr)^{-r}} \frac{k^{-rb}}{kr} = \frac{r^{r-1}\,k^{-br+r+1}}{\Gamma(r)\,(b\log(k))} \to 0$$

as $k \to 0$ because $b > (r+1)\,/\,r$. $\quad\square$

**Lemma 4.** *Let $b > 0$ and $\epsilon \in (0,1)$, as well as $k > 1$ and $n \in \{1, 2, \ldots\}$. If $(b\log(k))\,/\,k < 1$ and $X \sim Bin\,(n, (b\log(k))\,/\,k)$, then $P(X \geq n\epsilon) \leq (1 + b\log(k))^n\,/\,k^{n\epsilon}$.*

*Proof.* Let $Z_1, \ldots, Z_n \overset{\text{iid}}{\sim} Bern\,((b\log(k))\,/\,k)$. Because $x \mapsto k^x$ is strictly increasing,

$$P(X \geq n\epsilon) = P(k^X \geq k^{n\epsilon}) \leq \frac{\mathbb{E}[k^X]}{k^{n\epsilon}} = \frac{\prod_{i=1}^{n} \mathbb{E}[k^{Z_i}]}{k^{n\epsilon}} \leq \frac{(1 + b\log(k))^n}{k^{n\epsilon}}$$

via Markov's inequality. $\quad\square$

*Proof of lemma 2.* First, let $\epsilon \in (0,1)$. Next, let $b = (r+2)\,/\,r$ and choose $k^* \in \{2, 3, \ldots\}$ to be sufficiently large that $(1 + b\log(k))\,/\,k^\epsilon < 1$ and $(b\log(k))\,/\,k < \epsilon$ for all $k \geq k^*$. Then, for $k = 1, 2, \ldots, k^* - 1$, define $c_k = k$, and, finally, for $k = k^*, k^* + 1, \ldots$, define

$$c_k = k^{-1}(1 + b\log(k))^{2/\epsilon} + k\, P\left(\theta_k \geq \frac{b\log(k)}{k}\right),$$

where $\theta_k \sim \text{Beta}\left(r, (k-1) r\right)$.

Via lemma 3, $c_k \to 0$ as $k \to \infty$. Trivially, for $k < k^*$, $k\, P(N_1 \,/\, n \geq \epsilon \,|\, K = k, N = n) \leq k = c_k$.

Let $k \geq k^*$ and suppose that $n \geq 2\,/\,\epsilon$. Via a straightforward calculation, we can show that $N_1 \,|\, K = k, N = n \sim \text{BetaBin}\left(n, r, (k-1) r\right)$. (This follows from the fact that if $Y \sim \text{NegBin}\left(r, p\right)$ and, independently, $Z \sim \text{NegBin}\left(r', p\right)$, then $Y \,|\, (Y + Z) = n \sim \text{BetaBin}\left(n, r, r'\right)$.) Therefore, if we define $\theta \sim \text{Beta}\left(r, (k-1) r\right)$, $X \,|\, \theta \sim \text{Bin}\left(n, \theta\right)$, and $a = (b \log (k))\,/\,k$, then we have

$$k\, P(N_1 \,/\, n \geq \epsilon \,|\, K = k, N = n) = k\, P(X \geq n\epsilon)$$
$$= k\, P(X \geq n\epsilon, \theta < a) + k\, P(X \geq n\epsilon, \theta \geq a).$$

However, $k\, P(X \geq n\epsilon, \theta \geq a) \leq k\, P(\theta \geq a) = k\, P\left(\theta_k \geq \frac{b \log (k)}{k}\right)$. To handle the first term, we note that as a function of $\theta$, $P(X = x \,|\, \theta)$ is nondecreasing on $(0, \epsilon)$ whenever $x\,/\,n \geq \epsilon$ because $\frac{\mathrm{d}P(X=x\,|\,\theta)}{\mathrm{d}\theta} = \binom{n}{x} \theta^{x-1} (1-\theta)^{n-x-1} (x - n\theta)$. Therefore, $P(X \geq n\epsilon \,|\, \theta) = \sum_{x \geq n\epsilon} P(X = x \,|\, \theta)$ is nondecreasing on $(0, \epsilon)$. Finally, because our choice of $k^*$ means that $a \in (0, \epsilon)$,

$$k\, P(X \geq n\epsilon, \theta < a)$$
$$= k \int_0^a P(X \geq n\epsilon \,|\, \theta)\, P(\theta)\, \mathrm{d}\theta \leq k\, P(X \geq n\epsilon \,|\, \theta = a)$$
$$\leq k\, (1 + b \log (k))^n \,/\, k^{n\epsilon} = k \left(\frac{1 + b \log (k)}{k^\epsilon}\right)^n$$
$$\leq k \left(\frac{1 + b \log (k)}{k^\epsilon}\right)^{2/\epsilon} = k^{-1} (1 + b \log (k))^{2/\epsilon},$$

where the second inequality is via lemma 4 and the third inequality holds because $n \geq 2\,/\,\epsilon$ and $(1 + b \log (k))\,/\,k^\epsilon < 1$ because of our choice of $k^*$. Thus, $k\, P(N_1 \,/\, n \geq \epsilon \,|\, K = k, N = n) \leq c_k$. $\quad\square$

This completes the proof of theorem 1.

## C   The Chaperones Algorithm

For large data sets with many small clusters, standard Gibbs sampling algorithms (such as the one outlined in section 3) are too slow. In this appendix, we therefore propose a new Gibbs-type sampling algorithm, which we call the chaperones algorithm. This algorithm is inspired by existing split–merge Markov chain sampling algorithms [1, 2, 3]; however, it is simpler, more efficient, and—most importantly—likely exhibits better mixing properties when there are many small clusters.

In a standard Gibbs sampling algorithm, each iteration involves reassigning each data point $x_n$ for $n = 1, \ldots, N$ to an existing cluster or to a new cluster by drawing a sample from $P(C_N \,|\, N, C_N \setminus n, x_1, \ldots, x_N)$. When the number of clusters is large, this step can be inefficient because the probability that element $n$ will be reassigned to a given cluster will, for most clusters, be extremely small.

The chaperones algorithm focuses on reassignments that have higher probabilities. If $c_n \in C_N$ denotes the cluster containing data point $x_n$, then each iteration consists of the following steps:

1. Randomly choose two chaperones, $i, j \in \{1, \ldots, N\}$ from a distribution $P(i, j \,|\, x_1, \ldots, x_N)$ where the probability of $i$ and $j$ given $x_1, \ldots, x_N$ is greater than zero for all $i \neq j$. This distribution must be independent of the current state of the Markov chain $C_N$; however, crucially, it may depend on the observed data points $x_1, \ldots, x_N$.

2. Reassign each $x_n \in c_i \cup c_j$ by sampling from $P(C_N \,|\, N, C_N \setminus n, c_i \cup c_j, x_1, \ldots, x_N)$.

In step 2, we condition on the current partition of all data points except $x_n$, as in a standard Gibbs sampling algorithm, but we also force the set of data points in $c_i \cup c_j$ to remain unchanged—i.e., $x_n$ must remain in the same cluster as at least one of the chaperones. (If $n$ is a chaperone, then this requirement is always satisfied.) In other words, we view the non-chaperone data points in $c_i \cup c_j$ as "children" who must remain with a chaperone at all times. Step 2 is almost identical to the restricted Gibbs moves found in existing split–merge algorithms, except that the chaperones $i$ and $j$ can also change clusters, provided they do not abandon any of their children. Splits and merges can therefore

occur during step 2: splits occur when one chaperone leaves to form its own cluster; merges occur when one chaperone, belonging to a singleton cluster, then joins the other chaperone's cluster.

The chaperones algorithm can be justified as follows: For any fixed pair of chaperones $(i, j)$, step 2 is a sequence of Gibbs-type moves and therefore has the correct stationary distribution. Randomly choosing the chaperones in step 1 amounts to a random move, so, taken together, steps 1 and 2 also have the correct stationary distribution (see, e.g., [4], sections 2.2 and 2.4). To guarantee irreducibility, we start by assuming that $P(x_1, \ldots, x_N \,|\, C_N) \, P(C_N) > 0$ for any $C_N$ and by letting $C'_N$ denote the partition of $N$ in which every element belongs to a singleton cluster. Then, starting from any partition $C_N$, it is easy to check that there is a positive probability of reaching $C'_N$ (and vice versa) in finitely many iterations; this depends on the assumption that $P(i, j \,|\, x_1, \ldots, x_N) > 0$ for all $i \neq j$. Aperiodicity is also easily verified since the probability of staying in the same state is positive.

The main advantage of the chaperones algorithm is that it can exhibit better mixing properties than existing sampling algorithms. If the distribution $P(i, j \,|\, x_1, \ldots, x_N)$ is designed so that $x_i$ and $x_j$ tend to be similar, then the algorithm will tend to consider reassignments that have a relatively high probability. In addition, the algorithm is easier to implement and more efficient than existing split–merge algorithms because it uses Gibbs-type moves, rather than Metropolis-within-Gibbs moves.

## D  The Syria2000 and SyriaSizes Data Sets

Figure 1: Cluster size distributions for the Syria2000 (left) and SyriaSizes (right) data sets.

## Footnotes

[2] We have not truncated the negative binomial distributions, so this is a minor variant the NBNB model.