[Reviews · NeurIPS 2016]

Reviewer 1

Summary

The paper presents an alternative prior for Bayesian non-parametric clustering. The prior is a better fit for applications where the size of data in each cluster doesn't grow linearly with the number of data points (i.e. rich gets richer assumption as in DP and related priors is no longer valid). The authors explicitly modeled the distribution of cluster sizes in the new process and provided two incarnations of the process using different forms of distributions over cluster sizes. They gave a Gibbs-sampling algorithm as well as a split-merge algorithm in the appendix. Comparisons were given to DP and PYP over various datasets.

Qualitative Assessment

The authors addressed an important problem in BNP clustering and gave an interesting prior. However, the paper have the following weaknesses: 1- As the authors mentioned in line 64, [13] had a take on the same problem by defining a uniform prior rather than the rich gets richer prior implicit in DP and PYP. They showed that their prior produces larger number of clusters than DP. Unfortunately, the authors didn't compare with [13] nor discussed in any details, beyond a passing mention in line 64, how their work relates and differs from [13] in terms of the properties of the resulting prior. 2- The authors didn't discuss in details what are the implications of sacrificing consistency to achieve the micro-clustering property and how this compared to sacrificing exchangeability as in [13]? 3- The experimental results are not conclusive. Are any of the improvements statistically significant? I don't see a radical dominating performance across all measures of the new processes over exciting ones. 4- Writing is a bit hard to follow even for someone familiar with the area. Lots of symbols! Could use some work and simplifications.

Confidence in this Review

2-Confident (read it all; understood it all reasonably well)


Reviewer 2

Summary

The paper proposes a distribution over partitions of integers that supports micro-clustering. Specifically, the paper shows that that for existing infinitely exchangeable clustering models, such as Dirichlet Process (DP) and Pitman-Yor Process (PYP) mixture models, the cluster sizes grow linearly with the number of data points N. For applications such as entity resolution, the paper defines the micro-clustering property, where the ratio of the cluster size and number of data points N goes to 0 as N goes to infinity. The paper proposes a general distribution framework over partitions of integers that satisfies this micro-clustering property. This is done by first sampling the number of clusters from a distribution with positive integers as support, and then explicitly sampling each of the cluster sizes from a distribution over cluster sizes again with positive integers as support. This model achieves the micro-clustering property by sacrificing consistency of marginals, while preserving exchangeability. The paper then proposes two specific instances of this framework. The first uses the negative binomial for both distributions. The second uses a Dirichlet with an infinite dimensional base distribution for the distribution over cluster sizes to provide more flexibility for large datasets. Reseating algorithms similar to the Chinese Restaurant Process and the Pitman-Yor Process are provided for both models. Making use of the exchangeability property, sampling based algorithms are used for inference in both models. Experiments over 4 semi-synthetic datasets are used to illustrate that the proposed models outperform models without the micro-clustering property (DP and PYP) for the entity resolution task.

Qualitative Assessment

The following are the main strengths of the paper. + It points out and defines an important property of cluster sizes that existing infinitely exchangeable clustering models do not satisfy. There could be many applications, including and not limited to entity resolution, that require this property to be satisfied. + It proposes a framework for defining infinitely exchangeable clustering models that satisfy this micro-clustering property, and analyzes why the DP mixture model is an unsatisfactory instance of this class. It then proposes two specific and interesting instances of this class using specific distributions for the number of clusters and cluster sizes and derives reseating algorithms for these instances. + Detailed experimental results over semi-synthetic data illustrate usefulness of the proposed models to some extent for the entity resolution task. Experiments in the supplement show that draws from the proposed model satisfy the micro-clustering property. (This should be moved to the main paper.) But the paper also has certain deficiencies, some of which are probably fixable. - The paper points out towards the beginning that it preserves exchangeability and sacrifices consistency of the marginal distributions to achieve the micro-clustering property. While I can see the benefits of preserving exchangeability (e.g. in designing sampling based inference algorithms), the price of sacrificing consistency is not clear. The paper has a vague statement in line 109: "It does not produce an exact sample if used to incrementally construct a partition". What does this mean? Is this related to the consistency issue? - While the paper has empirical comparisons with the Pitman-Yor Process, there is no theoretical analysis of the differences in the reseating algorithms. I can understand that the goal is to have significantly heavier tails in the cluster size distribution compared to the DP / CRP. The PYP addresses this by making the probability of picking a new table proportional to the current number of clusters. The reseating algorithm for both proposed models looks similar. Which aspect of the proposed reseating helps it to satisfy the micro-clustering property while the PYP does not? - The paper also does not satisfactorily analyze the relationship between the two proposed models. The Negative Binomial in the first model is replaced by a Dirichlet with an infinite dimensional base distribution in the second. In the second case, I could think of the cluster size probabilities as drawn from a GEM / Poisson-Dirichlet with two parameters. Would this also become too restricted? Or is the difference due to two different priors over infinite dimensional multinomial parameters, e.g. Negative Binomial and GEM? - Early on, the paper mentions related work (the Uniform Process) that achieves the same goal differently: by sacrificing exchangeability instead of consistency. This work is never mentioned later in the paper. Some discussion of this earlier paper is necessary to understand the contributions of the proposed framework. - The experiments are not convincing enough for appreciating the usefulness of the proposed models for the entity resolution task. Judging by the error rates for all the models, the task for the first two datasets seem too simple for these datasets to be significant. For the other two datasets (which the paper claims are too hard), the two baseline models seem to be doing better. Some less important comments: - Where is alpha in the reseating algorithm in lines 131-133? - If the authors feel that the Chaperones algorithm is significant, they need to describe it at least at a high level in the main paper. Mentioning it just by name in the main paper is not helpful. - The flow of the paper could be changed for better understanding. Lines 112-115 and 134-139 could be moved to the inference section, and lines 116-118 and 140-144 could be moved to the experiments section. - The authors should reconsider the name of the proposed model. They should at least change it to the Flexible Micro-clustering Model instead of the Flexible Model for Micro-clustering model. - In Eqn 3 and line 85, |C|_N should be |C_N|. In line 84, cluster |c| should be cluster c. - Eqn 5 should have a,q in the conditional. Eqn 7 should have a,q,alpha in the conditional. - What does "K is one observation" in line 122 mean? - In Eqn 11, please explain that theta_lk is the lth feature for the kth partition. - (The partition features of) One synthetic dataset can be imagined as one draw from (the entity features of) the corresponding real dataset. Are experimental results reported over a single random synthetic dataset or averaged over multiple samples? Though the noise parameters may be set to achieve low noise in expectation, it is possible to get high noise in a single random draw. - What is the difference between the two Syria datasets? - How are the error rates defined accounting for the label permutation issue? Are these defined over pairs of data points? - Some sentences contain unnecessary repetitions, e.g. in lines 150 and 214-216

Confidence in this Review

3-Expert (read the paper in detail, know the area, quite certain of my opinion)


Reviewer 3

Summary

The authors propose a Bayesian nonparametric method for clustering data where the size of clusters grows like o(n) rather than O(n/log(n)), which is assumed in most common models like a Dirichlet process. They propose a generative model where (1) the number of clusters is drawn from a distribution, and then (2) the number of items in each cluster is drawn from a second distribution. Submodels are proposed based on the prior distributions for (1) and (2), including a negative binomial/negative binomial model (NBNB) and a negative binomial/Dirichlet model (NBD). These are tested on survey data.

Qualitative Assessment

Bayesian clustering models for small cluster sizes is an interesting (and difficult) problem as many real world applications have many small clusters. Technical quality: The biggest issue with the paper is model incoherence, as the authors noted. Incoherence means that marginal distributions generated using the full data set do no necessarily coincide with the distribution using a subset of the data. This model maintains data exchangeability at the cost of incoherence. The authors did not make clear the implications of removing exchangeability vs incoherence for microclustering. For what it is worth, non-exchangeability seems like a more reasonable modeling assumption with microclustering, so please provide motivating counter examples. I am not in the camp that incoherence is always a fatal flaw, but it does need to be approached more carefully. In which situations can this model be used? In which situations can it not be used? What sorts of problems does it create when used as a generative model? Other areas: the paper was fairly clear, methods were well-implemented, and the potential usefulness is moderate to good.

Confidence in this Review

2-Confident (read it all; understood it all reasonably well)


Reviewer 4

Summary

The authors propose the notion of micro clustering, where models exhibit the micro clustering property if cluster sizes grow sub linearly with the size of the data. The need for models for micro clustering is well motivated by the problem of entity resolution, which involves identifying duplicate records in noisy databases. The authors show that the commonly used mixture models for clustering do not satisfied the micro clustering property and introduce a novel class of partition models. They focus on two specific models within this class, and show empirically that the micro clustering property is satisfied. In addition, they develop a Gibbs sampling algorithm for posterior inference. The authors compare the two proposed models with the popular Dirichlet process and Pitman-Yor process mixture models in four realistic simulated datasets. They find that in simulated de-duplication scenarios, the proposed models perform better but in noisy datasets with a small number of features, performance is worse and similar across the four models.

Qualitative Assessment

This is an interesting and well-written paper and the proposed models are well motivated by the problem of entity resolution. My main concern is with the introduced class of "Flexible models for Microclustering" (FMMC). In particular, I expected the authors to prove that this class of models satisfies the micro clustering property, and there is no such proof in the manuscript. Moreover, the authors state that the partition model induced by the DP is a special case in lines 91-94 (although I find this confusing as \mu and \kappa depend on the sample size N, which is not the case in the definition in eq (2)) but also discuss how the DP does not satisfy the micro clustering property. Thus, the name for this class of models seems inappropriate, as models within the class may not satisfy the micro clustering property. This also brings into question the motivation for introducing this class of models; I quite liked the motivation to develop micro clustering models for entity resolution but as previously stated the introduced class of models may not satisfy the micro clustering property. The paper could be strengthen by a Theorem that states the conditions of \mu and \kappa needed for the micro clustering property to hold. In addition, I was also disappointed that for two proposed models, the NBNB and NBD, the micro clustering property was only shown to hold empirically. The paper could be strengthen by showing that the choice of \mu and \kappa for these two models satisfies the Theorem. Comments: 1. pg.3 eq. 3 and line 85: \kappa_{|C|_N} should be \kappa_{|C_N|}. 2. pg.3 line 88: We now present two flexible FMMC (i.e. remove more) 3. pg. 5 line 200-201: should say posterior expected number of cluster and posterior expected value of \delta_l (I assume you are using the posterior expectation to estimate these quantities?) 4. pg. 5 line 201: the posterior expected value of \delta_l reported in the tables is a single number. I expected $L$ different values. Are you assuming a common \delta across l or taking an average? Please clarify. 5. pg. 6 Figure 1: please label panels a,b,c,d,

Confidence in this Review

2-Confident (read it all; understood it all reasonably well)


Reviewer 5

Summary

The paper proposes two models for microclustering. Usual tools for model-based clustering rely on latent (allocation) variables, such as implied by the Dirichlet (or related) process mixture models, and they lead to linear growth of cluster sizes. Here, sublinear cluster size growth is obtained (not theoretically but empirically by simulation) by dropping a "consistence in distribution" assumption.

Qualitative Assessment

The paper is very interesting, well written, and fits nicely into a conference like NIPS as it introduces novel methodology (model & algorithm) to tackle important applications (entity resolution). I would be interested to see some more insight/discussion on the microclustering property, or pointer to ongoing research about it, either in the text or in Appendix B. For instance, the log-log plot of Fig 2 in Appendix B suggests a power law decay of M_N/N with an exponent around 0.8. I feel it would be interesting to comment about that. BTW, both plots in Fig 2 are identical (ie same Monte Carlo error), they should be updated. Also, correct caption "top/bottom". Minor comments: - correct subscript N in eqn (3) as well as line 85 - page 4, footnote 3: shouldn't the partitions C_N be ordered for zeta to be well defined? of course it is just a matter of notation, but I would say that they should be denoted eg ({1, 3, 4}, {2, 5}, {6}) instead of {{1, 3, 4}, {2, 5}, {6}} - lines 81 & 151: use consistent notation for partition space, for instance \mathcal{C}_N - line 287, add that K = |C_N| - after line 298: use the same Gamma parameters than those introduced in main text line 100 - line 303, displayed eqn: correct c \in C_N

Confidence in this Review

3-Expert (read the paper in detail, know the area, quite certain of my opinion)


Reviewer 6

Summary

The paper considers clustering task with models that yield clusters whose size grow sublinearly (instead of linear) with the total number of data points. The authors define the microclustering property to address this requirement and propose two new models (the NBNB model and the NBD model) that exhibit this property.

Qualitative Assessment

The microclustering property defined in this paper provides a way to characterize models that require clusters whose sizes grow sublinearly, including entity resolution. The paper propose NBNB and NBD model that exhibit this property and compare their performance using real datasets. Besides lacking of theoretical justification, the paper overall may have large impact for further research on clustering models.

Confidence in this Review

1-Less confident (might not have understood significant parts)